# Unveiling the double-peak structure of quantum oscillations in the specific heat

Zhuo Yang [1] ✉, Benoît Fauqué [2], Toshihiro Nomura[1], Takashi Shitaokoshi[1], Sunghoon Kim [3], Debanjan Chowdhury[3], Zuzana Pribulová [4], Jozef Kačmarčík [4], Alexandre Pourret[5], Georg Knebel [5], Dai Aoki[6], Thierry Klein[7], Duncan K. Maude[8], Christophe Marcenat[5] & Yoshimitsu Kohama [1]

Quantum oscillation phenomenon is an essential tool to understand the electronic structure of quantum matter. Here we report a systematic study of quantum oscillations in the electronic specific heat $C_{el}$ in natural graphite. We show that the crossing of a single spin Landau level and the Fermi energy give rise to a double-peak structure, in striking contrast to the single peak expected from Lifshitz-Kosevich theory. Intriguingly, the double-peak structure is predicted by the kernel term for $C_{el}/T$ in the free electron theory. The $C_{el}/T$ represents a spectroscopic tuning fork of width $4.8k_BT$ which can be tuned at will to resonance. Using a coincidence method, the double-peak structure can be used to accurately determine the Landé $g$-factors of quantum materials. More generally, the tuning fork can be used to reveal any peak in fermionic density of states tuned by magnetic field, such as Lifshitz transition in heavy-fermion compounds.

Oscillations of the physical properties of materials with magnetic fields are powerful tools to reveal the electronic properties of quantum matter. They range from Aharonov–Bohm oscillations[1] in mesoscopic rings, which provide a direct measure of the electron coherence, to quantum oscillations which provide a sensitive and incisive probe of the Fermi surface. In the latter case, with increasing the magnetic field, the Landau quantization of the carrier motion gives rise to a series of quantized singularities in the density of states (DOS) that cross the Fermi level, resulting in the oscillatory behavior of various physical quantities, such as resistivity (Shubnikov–de Haas effect), magnetic susceptibility (de Haas–van Alphen effect), thermopower and specific heat.

Lifshitz–Kosevich (LK) theory has been widely used to describe these oscillatory phenomena[2–4], notably to extract parameters such as the effective mass and Landé $g$-factor. Although the theory is remarkably successful in describing quantum oscillations in metals over a wide range of magnetic fields and temperatures, there is

growing evidence to suggest that experiment often deviates from the predicted LK behavior[5–9]. At high magnetic fields, the oscillatory magnetoresistance[5–7], magnetization[9], and thermopower[8] exhibit a clear departure from LK theory when the systems are pushed towards the quantum limit. It is natural to expect that a similar departure is also observed in specific heat. However, the oscillatory behavior of the specific heat in the quantum limit has yet to be fully explored. In this respect, graphite, in which the quantum limit is reached already at fields as low as 7 T[10], is almost an ideal system for this purpose.

In this study, we report the quantum oscillations of specific heat in natural graphite with temperatures down to 90 mK. Intriguingly, as the field increases and the system approaches the quantum limit, a characteristic double-peak structure appears in the specific heat for magnetic fields corresponding to the expected crossing of an individual spin Landau level and the Fermi energy. This result is in striking contrast to the single peak feature predicted in LK theory for the quantum

[1]Institute for Solid State Physics, The University of Tokyo, Kashiwa, Chiba 277-8581, Japan. [2]JEIP, USR 3573 CNRS, Collège de France, PSL Research University, 11, Place Marcelin Berthelot, 75231 Paris Cedex 05, France. [3]Department of Physics, Cornell University, Ithaca, NY 14853, USA. [4]Centre of Low Temperature Physics, Institute of Experimental Physics, Slovak Academy of Sciences, Watsonova 47, SK-04001 Košice, Slovakia. [5]Univ. Grenoble Alpes, CEA, Grenoble INP, IRIG, PHELIQS, 38000 Grenoble, France. [6]Institute for Materials Research, Tohoku University, Oarai, Ibaraki 311-1313, Japan. [7]Univ. Grenoble Alpes, CNRS, Institut Néel, 38000 Grenoble, France. [8]Laboratoire National des Champs Magnétiques Intenses, CNRS-UGA-UPS-INSA, 143 avenue de Rangueil, 31400 Toulouse, France. ✉e-mail: zhuo.yang@issp.u-tokyo.ac.jp

oscillations of specific heat, which is widely used in the literature[11-13] (see also Supplementary Note 1). The double-peak structure, which unexpectedly vanishes as $T \to 0$, occurs when a narrow Landau level crosses the thermally broadened edge of the Fermi–Dirac distribution in the vicinity of the Fermi energy. We demonstrate that the double-peak structure in the oscillatory specific heat originates from the kernel term in the detailed functional form of the free electron theory expression for the specific heat[14]. A quantitative understanding of the double-peak structure is achieved by the comparison of a DOS model and the Slonczewski–Weiss–McCure (SWM) tight-binding Hamiltonian for graphite[15,16]. Using graphite as an example, we demonstrate that the double-peak structure provides a new way to accurately determine the g-factor of charge carriers without any assumptions concerning the Landau index or Fermi energy shift, and it can also be extended to other Dirac materials which is crucial in the determination of the Berry phase. Furthermore, the double-peak structure detected here is not restricted to $C_{el}$ in the presence of Landau quantization. It can occur in other probes related to specific heat, such as thermal conductivity, and in any system where a fermionic sea is tuned by the magnetic field such as a Lifshitz transition or in frustrated magnetic materials with fermionic-like excitations.

## Results
### Experimental results
When a Landau level crosses the Fermi energy, the occupation of the Landau level changes rapidly, inducing large changes in the entropy of the system, which can be probed using thermodynamic measurements. The magnetocaloric effect (MCE) measures the sample temperature as a function of the applied magnetic field under quasi-adiabatic conditions. In this case, the absolute value of entropy is roughly proportional to the reciprocal sample temperature. To follow the evolution of the entropy in our graphite sample, we show in Fig. 1a the measured reciprocal sample temperature ($1/T$) as a function of the magnetic field taken at an initial temperature of 0.7 K. The field was applied along the c-axis of the graphite crystal for all the measurements presented in this paper. The entropy is proportional to the logarithm of the number of states within the Fermi edge, and therefore shows a maximum when a Landau level is located at the Fermi level, resulting in a series of well-defined single peaks (Supplementary Note 7) in the reciprocal sample temperature labeled as $N_{e/h}^{\pm}$ in Fig. 1a. Here, $N$ is the Landau index, e/h indicates if the Landau level originates from the electron or hole pocket, and $\pm$ indicate the spin up/down levels. For better comparison, Fig. 1b shows background removed magnetoresistance $\Delta R_{xx}$ on natural graphite at 0.5 K.

These results are in stark contrast to the electronic specific heat divided by temperature $C_{el}/T$ which is proportional to the temperature derivative of entropy. The $C_{el}/T$ of the graphite sample taken at a similar temperature ($T = 0.5$ K) is shown as a function of the magnetic field in Fig. 1c. The electronic specific heat $C_{el}$ was obtained by subtracting the phonon contribution from the total specific heat (Supplementary Note 2). Crucially, when low-index Landau levels ($N_{e/h} < 3$) cross the Fermi energy, $C_{el}/T$ exhibits a series of double-peak structure,

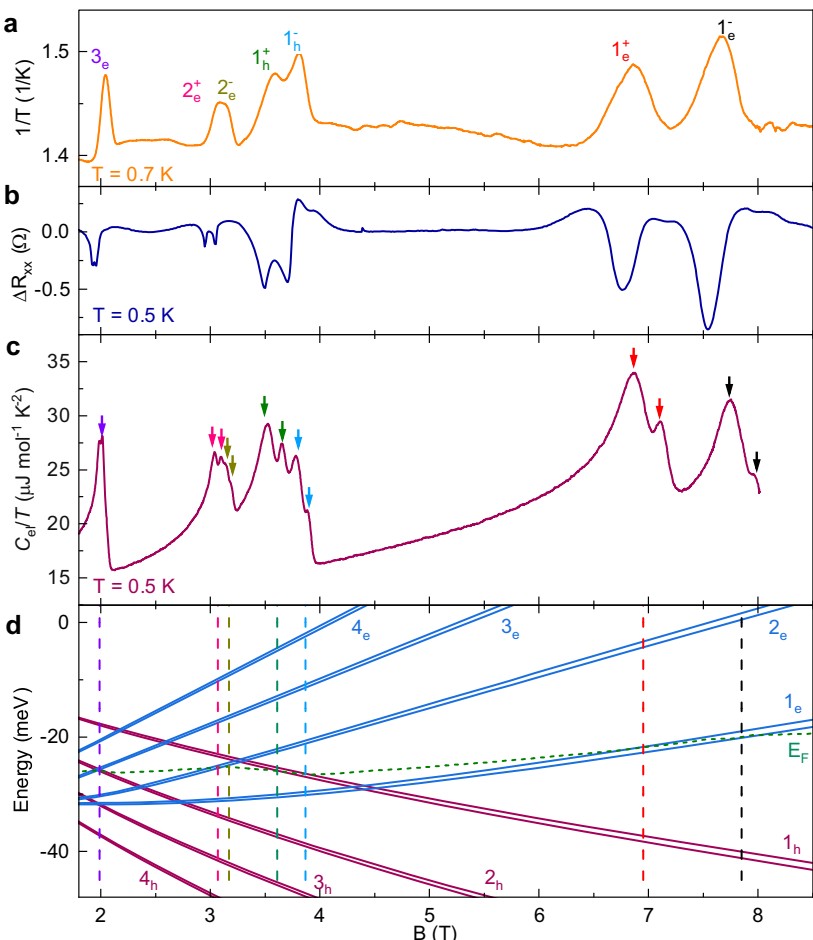

**Fig. 1 | Comparison of quantum oscillations in MCE, resistivity and specific heat. a** Reciprocal temperature ($1/T$) of graphite as a function of applied magnetic field in a quasi-adiabatic condition measured at initial temperature $T = 0.7$ K. **b** Background removed resistance $\Delta R_{xx}$ as a function of magnetic field at $T = 0.5$ K for natural graphite. **c** Field sweep electronic specific heat divided by temperature $C_{el}/T$ in graphite as a function of magnetic field at $T = 0.5$ K. **d** Electron (blue) and hole (red) Landau levels calculated within SWM-model for $B \geq 1.8$ T.

as indicated by the double arrows in Fig. 1c. Our observations demonstrate that as we approach the quantum limit, Landau levels crossing the Fermi energy give rise to single features in MCE and magnetoresistance, while simultaneously a novel double-peak structure is observed in the specific heat $C_{el}/T$. To verify that the double-peak structure in $C_{el}/T$ is an intrinsic effect, we have measured $C_{el}/T$ versus magnetic field for three different samples, together with an angle-dependence $C_{el}/T$ (Supplementary Note 3). The double-peak structure in $C_{el}/T$ shows good reproducibility and follows the expected (for graphite) quasi-2D behavior in the magnetic field, allowing us to conclude that the double-peak structure is an intrinsic effect.

Strikingly, the magnetic field splitting $\Delta B = B_2 - B_1$ of the double-peak structure in $C_{el}/T$ is strongly temperature dependent and vanishes as $T \to 0$. Figure 2a shows the double-peak structure in $C_{el}/T$ as a function of the magnetic field for the $N = 1_h^{\pm}$ Landau levels measured for different temperatures from 90 mK to 1.5 K. The $C_{el}/T$ curves are vertically shifted for clarity. Symbols indicate the peak positions (corresponding to fields $B_1$ and $B_2$) for $1_h^+$ and $1_h^-$ levels, respectively. At high temperatures, the splitting is clearly resolved. At lower temperatures, the splitting decreases, and the double-peak structure eventually merges into a single peak at $T = 90$ mK. To analyze the evolution of the splitting, we plot the magnetic field position of the double-peak structure as a function of temperature in Fig. 2b for various spin up/spin down hole and electron Landau levels. The peak positions $B_1$, $B_2$ scale linearly with the temperature and the extrapolated splitting vanishes at $T = 0$ K for all Landau levels. The $T$-linear dependence of the peak positions $B_1$, $B_2$ is a characteristic feature for the double-peak structure presented in this study.

The SWM Hamiltonian[15,16] with its seven tight binding parameters $\gamma_0, ..., \gamma_5, \Delta$ provides a remarkably accurate description of the band structure of graphite[17,18]. In a first approach, we use the SWM model to understand the observed Landau level crossings with the Fermi energy. The Landau levels were calculated by finding the local extrema for each Landau band ($dE_N/dk_z = 0$), where a saw-tooth-like singularity in the DOS is located. Moreover, as we approach the quantum limit, the

movement of the Fermi energy to keep the charge neutrality is non-negligible, and inevitably influences the magnetic field at which a given Landau level crosses the Fermi level[18–20]. For this reason, the Fermi level movement was calculated based on the principle of charge neutrality, that is, the difference of the electron ($n$) and hole ($p$) carrier concentration is a constant: $n - p = n_0$. For the SWM parameters, we used the values which were fine-tuned to correctly reproduce de Haas-van Alphen measurements in natural graphite[21] (Supplementary Table 1). The calculated results are shown in Fig. 1d. Solid lines show the evolution of the lowest electron and hole Landau levels with the magnetic field, while the green dashed line shows the calculated evolution of the Fermi level. To facilitate the comparison of theory and experiment, we draw a series of vertical dashed lines in Fig. 1d to indicate magnetic fields corresponding to the crossing of electron/hole Landau levels with the Fermi energy. The positions of the dashed lines are in near-perfect agreement with the magnetic fields of the observed peaks in MCE and the double-peak structure in $C_{el}/T$.

## Discussion
### Origin of double-peak structure
In order to elucidate the origin of the double-peak structure in $C_{el}/T$ versus $B$, it is necessary to consider the exact form of the expression for the specific heat. For electronic quasiparticles, $C_{el}/T$ is given by[14],

$$C_{el}/T = k_B^2 \int_{-\infty}^{\infty} D(E)\left(-x^2 \frac{dF(x)}{dx}\right) dx, \qquad (1)$$

where $F(x) = 1/(1 + e^x)$, $x = E/k_B T$ and $k_B$ is the Boltzmann constant. The specific heat depends on the convolution of the Landau level DOS $D(E)$ and a kernel term $-x^2 dF(x)/dx$ which involves the first derivative of the Fermi-Dirac distribution function. The usual approximation, removing $D(E)$ from the integral, and replacing it with $D(E_F)$, to obtain the well-known formula $C_{el} = \frac{1}{3}\pi^2 D(E_F) k_B^2 T$[14], actually suppresses the double-peak structure in $C_{el}/T$[22]. As we will see, the double-peak structure in $C_{el}/T$ originates in the temperature-dependent splitting of the double

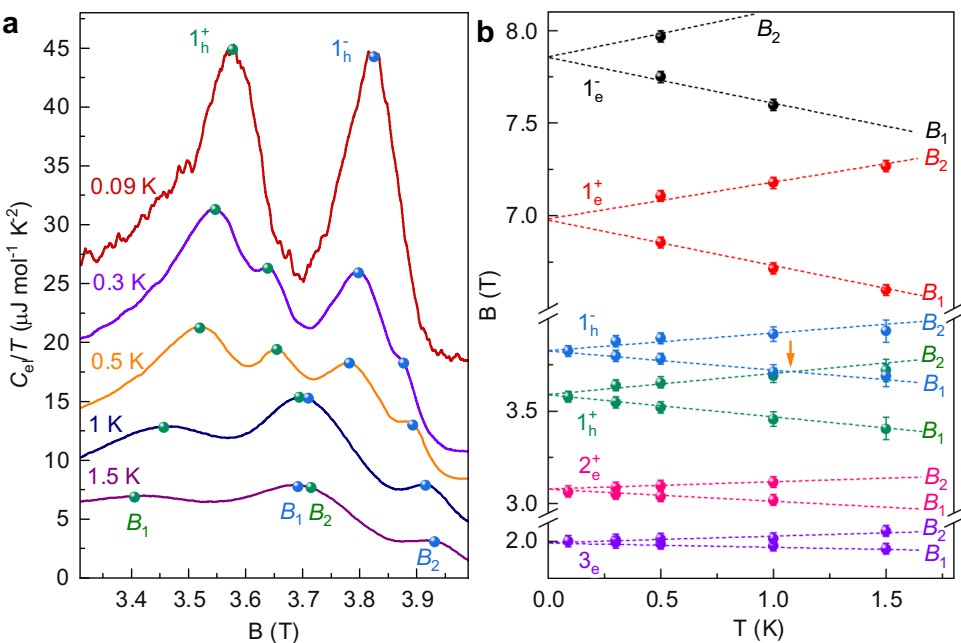

**Fig. 2 | Temperature dependence of the double-peak structures in $C_{el}/T$.**
**a** Electronic specific heat divided by temperature $C_{el}/T$ in graphite as a function of magnetic field in the vicinity of the $1_h^{\pm}$ Landau level/Fermi energy crossing. Measurements at different temperatures are vertically offset for clarity. At higher temperature each spin Landau level gives rise to a double-peak structure (peaks indicated by symbols) as it crosses the Fermi energy. **b** Measured magnetic field position of the double-peak structure $B_1$, $B_2$ as a function of temperature for various spin up/spin down electron and hole Landau levels. The error bars represent one standard deviation of uncertainty.

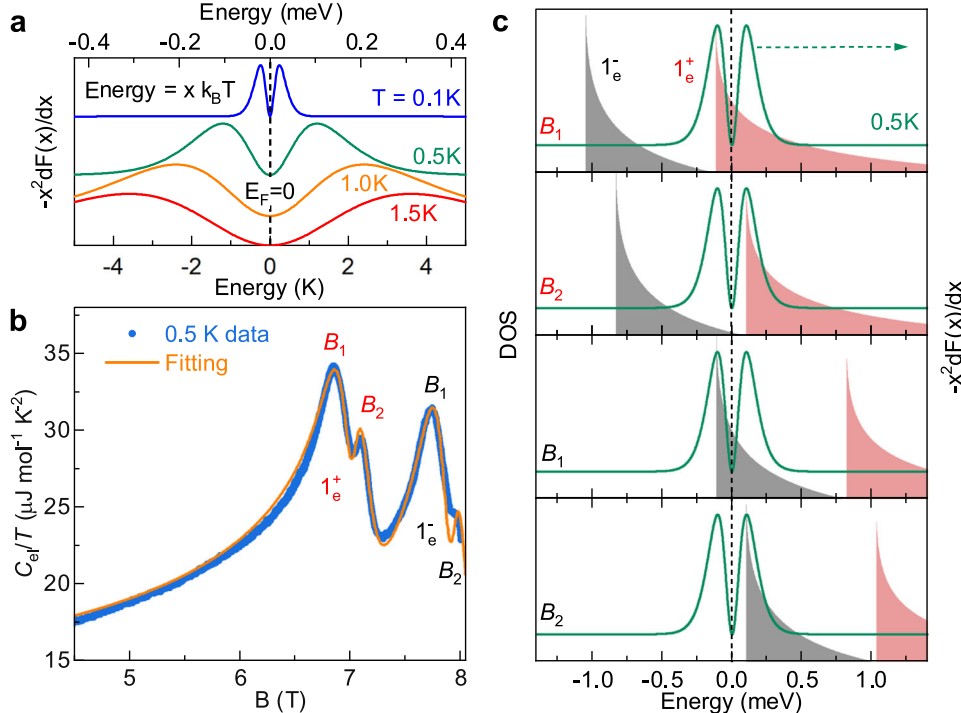

**Fig. 3 | Origin of the double-peak structure: The kernel term $-x^2 dF(x)/dx$. a** The kernel term $-x^2 dF(x)/dx$ (curves offset vertically for clarity) in the vicinity of the Fermi energy calculated at different temperatures and plotted versus $E = xk_BT$. For $-x^2 dF(x)/dx$ maxima occur at $x = \pm 2.4$ so that the splitting of the maxima is $\Delta E = 4.8 k_B T$. **b** Measured and calculated electronic specific heat divided by temperature $C_{el}/T$ in graphite as a function of magnetic field at $T = 0.5$ K. The exceptional quality of the fit, notably the position and amplitude of the double-peak structure, together with the characteristic asymmetric line-shape demonstrates the validity of our simple DOS model. **c** Schematic to show the origin of the double-peak structure in the left panel - we plot the DOS of the $1_e^\pm$ spin split Landau level at magnetic fields corresponding to crossing the maxima in $-x^2 dF(x)/dx$ calculated here for $T = 0.5$ K.

maxima in the kernel term $-x^2 dF(x)/dx$ (when plotted versus $E = xk_BT$). To illustrate this, in Fig. 3a we plot the kernel term $-x^2 dF(x)/dx$ in the vicinity of the Fermi energy ($x = 0$) at different temperatures. This function shows a distinctly non-monotonic behavior with maxima located at $x = \pm 2.4$. The maxima on either side of the Fermi energy occur at an energy $E = \pm 2.4 k_B T$ (Supplementary Note 8), so that the splitting of the maxima $\Delta E = 4.8 k_B T$ varies linearly with temperature and vanishes as $T \to 0$. Qualitatively, this exactly predicts the temperature dependence exhibited by the double-peak structure in $C_{el}/T$. The double-peak structure in quantum oscillations was also predicted in earlier theoretical calculations using an explicit expression for the specific heat[23]; however, a quantitative comparison between experiment and theory is still missing.

To obtain a quantitative comparison, we use a model DOS, calculating the specific heat $C_{el}/T$ as the Landau level crosses the Fermi energy using Eq. (1). The shape of Landau quantized three-dimensional DOS is saw-tooth-like, resulting from the superposition of the quantized DOS of a two-dimensional system perpendicular to the field direction (delta function) and the density of states due to the dispersion along $k_z$ (DOS $\propto 1/\sqrt{E}$)[24]. We approximate the DOS for a single Landau level, with its "singularity" at $E = E_0$, by the following rigid expression for energies $E \geq E_0$,

$$D(E) = \frac{A}{1 + \sqrt{(E - E_0)/\Gamma}}, \qquad (2)$$

The one in the denominator prevents the unphysical (in a real system) divergence of the DOS which has a maximum amplitude of $A$ at $E = E_0$. The parameter $\Gamma$ is the full width at half maximum (FWHM) of the Landau level. For simplicity all energies are calculated relative to the Fermi energy $E_F = 0$. The position of the Landau level at a given magnetic field is $E_0 = (B - B_0) dE/dB$, $B_0$ is the magnetic field at which the

Landau level crosses the Fermi energy at $T = 0$. In this model, the magnetic field dependence of the Landau level energy $dE/dB$ measured relative to the Fermi energy is a fitting parameter, and thus includes the cyclotron energy and Zeeman energy, together with any movement of the Fermi energy in the magnetic field. We stress that the double-peak structure, is the result of a single spin Landau level crossing the Fermi energy. However, since the spin splitting is small, the spin up/down Landau levels cross $E_F$ in quick succession, generally producing quadruple peaks. In order to locally fit the $C_{el}/T$ data we use $D_{\uparrow\downarrow}(E) = D_\uparrow(E) + D_\downarrow(E)$ with $E_0^{\uparrow\downarrow} = (B - B_0^{\uparrow\downarrow}) dE/dB$. In this approximation, the spin split Landau level rigidly shifts through the Fermi energy i.e. the spin gap remains constant over the limited magnetic field range involved. This approximation is justified by the fact that the extracted $dE/dB$ values for a given spin up/down Landau level are quasi-identical, and for simplicity we force them to be identical in the final fit.

Figure 3b shows the magnetic field dependence of $C_{el}/T$ data in the magnetic field region where the $1_e^\pm$ spin Landau levels cross the Fermi energy, together with the results of the fit. The fitting parameters used are $\Gamma = 0.21$ meV, $B_0^\uparrow = 7.05$ T, $B_0^\downarrow = 7.80$ T, and $dE/dB = 1.04$ meV/T. Note that the FWHM $\Gamma$, obtained here by fitting $C_{el}/T$ versus $B$, is very close to the Landau level broadening $\Gamma_q = \hbar/\tau_q$ determined from the magnetic field for the onset of the Shubnikov-de Haas oscillations ($\omega_c \tau_q = 1$) in natural graphite at mK temperatures reported in a previous study[25] (see Supplementary Table 2). The calculated curve is in excellent agreement with the $C_{el}/T$ data, reproducing correctly the position, and the amplitude of each double-peak structure feature, together with the asymmetric line-shape, which naturally arises due to the asymmetric nature of the Landau level DOS. Figure 3c schematically shows the DOS for the $1_e^\pm$ spin-split Landau level used to calculate $C_{el}/T$ at the four magnetic fields corresponding to maxima in $C_{el}/T$. For comparison, we also plot the kernel term $-x^2 dF(x)/dx$ calculated for the measurement temperature of $T = 0.5$ K. The peaks in $C_{el}/T$ appear at

**Table 1 | Summary of the parameters obtained from the simple DOS model and the SWM-Hamiltonian close to where the Landau levels cross the Fermi energy**

| LL | | $1_e^+$ | $1_e^-$ | $2_e^+$ | $2_e^-$ | $3_e$ | $1_h^+$ | $1_h^-$ | units |
|---|---|---|---|---|---|---|---|---|---|
| DOS | $\Gamma$ | 0.21 | 0.21 | 0.18 | 0.18 | 0.08 | 0.16 | 0.16 | meV |
| | $B_0$ | 7.05 | 7.87 | 3.08 | 3.17 | 2.00 | 3.61 | 3.86 | T |
| | $dE/dB$ | 1.04 | 1.04 | 4.45 | 4.45 | 7.1 | 1.74 | 1.74 | meV/T |
| SWM | $S_N\!-\!S_F$ | 0.98 | 0.98 | 4.06 | 4.06 | 7.47 | 2.11 | 2.11 | meV/T |
| | $S_N$ | 2.99 | 2.99 | 5.30 | 5.30 | 7.47 | 4.45 | 4.45 | meV/T |
| | $S_F$ | 2.01 | 2.01 | 1.24 | 1.24 | 0 | 2.34 | 2.34 | meV/T |

The tight binding parameters of the SWM Hamiltonian can be found in Supplementary Table 1.

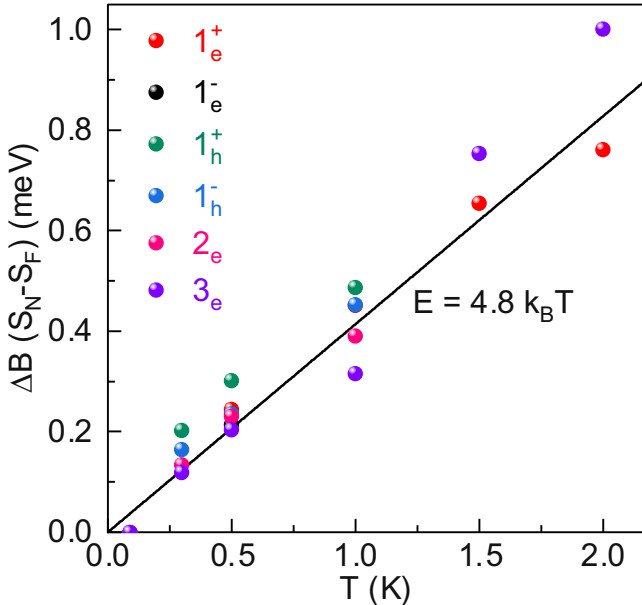

**Fig. 4 | Consistency between DOS model and SWM-Hamiltonian.** Temperature dependence of the energy through which the different Landau levels move $\Delta B (S_N\!-\!S_F)$ calculated using the movement of the Landau level relative to the Fermi energy extracted from the SWM model (see Table 1). All data collapses onto a single straight line through the origin. The solid line is the calculated $E = 4.8k_BT$ temperature-dependent splitting of the two maxima in $-x^2 dF(x)/dx$.

a certain magnetic field when the DOS peak is tuned to the maxima of the kernel term $-x^2 dF(x)/dx$.

In Table 1, we summarize parameters extracted from the simple DOS model for all the Landau levels. In order to compare the values of $dE/dB$ with the predictions of the SWM Hamiltonian, we calculate the slope of SWM Landau level energy with respect to the Fermi energy in the vicinity of the crossings i.e. $S_N\!-\!S_F$. Here, $S_N$ is the field dependence of the $N$-th SWM Landau level energy, while $S_F$ is the field dependence of the Fermi energy which originates from the charge neutrality condition. We see that the DOS model and SWM Landau level slopes agree to be within 10% ($dE/dB \simeq |S_N - S_F|$), which is very reasonable given the approximations involved. The good agreement between the predicted and experimental values lends further strong support to our model, and indicates that the double-peak structure in $C_{el}/T$ is a new way to access the Landau level dispersion.

A crucial test of our model for the origin of the double-peak structure is shown in Fig. 4. For each Landau level and each temperature we can compare the energy through which the Landau level moves (from field $B_1$ to $B_2$) with the energy separation of the maxima in $-x^2 dF(x)/dx$ which depends only on the temperature. The energy shift, as the magnetic field changes by $\Delta B = B_2 - B_1$, can be calculated

provided we know the slope of the Landau levels (movement relative to Fermi energy). In Fig. 4 we plot the energy shift of the Landau levels $\Delta B (S_N\!-\!S_F)$ versus temperature $T$ using the SWM values of $(S_N\!-\!S_F)$ summarized in Table 1. Plotted in this manner all of the data collapse onto a single straight line through the origin. The solid line is the expected splitting of the maxima in $-x^2 dF(x)/dx$, namely $E = 4.8k_BT$.

**Estimate of the $dE/dB$ from the double-peak structure**
The comparison between the DOS model and SWM model allows us to derive the following relation for a quantitive characterization of the double-peak structure:

$$\Delta B \frac{dE}{dB} = \Delta B|S_N - S_F| = 4.8k_BT, \qquad (3)$$

Intriguingly, Eq. (3) implies that the slope of the Landau level $dE/dB$ can be estimated based on the magnetic positions $B_1$, $B_2$ of the double peaks (note that $\Delta B = B_2 - B_1$). This is apparently useful for a new system with unknown shape of DOS peak, when the DOS model fitting is not applicable. Here, it is important to note that Eq. (3) is accurate provided the DOS peak is symmetric. However, in the case of asymmteric DOS peak, our simulations (Supplementary Fig. 9) show that $\Delta B(dE/dB)$ can be 10–20% larger than the $4.8k_BT$ splitting of $-x^2 dF(x)/dx$ depending on the Landau level width $\Gamma$. In addition, the asymmetric DOS peak also induces deviation between the peak position in $1/T$ and the center of double-peak structure (Supplementary Fig. 10).

As $T \to 0$, we expect the double-peak structure to merge into a single peak (as seen in 90 mK data in Fig. 2a), when the splitting (4.8 $k_BT$) of maxima of kernel term $-x^2 dF(x)/dx$ is smaller than linewidth of Landau level DOS. Due to the highly asymmetric nature of the Landau level DOS, this condition is fulfilled when the splitting of the maxima $4.8k_BT \simeq \Gamma/2$, i.e. half the FWHM of $D(E)$. Applying this condition, the values of $\Gamma$ extracted from the simple DOS model in Table 1 provide a reasonable estimate of the temperature below which the double-peak structure is quenched in the experimental $C_{el}/T$ data. For example, the double-peak structure disappears between 0.3 K and 0.09 K for the $1_h$ Landau level in Fig. 2, while the predicted quench temperature $\Gamma/9.6k_B \simeq 0.2$ K.

**Estimate of the $g$-factor from the double-peak structure**
In general, to extract the $g$-factor using techniques such as SdHs, dHvA, MCE, etc, one has to know the Landau index (orbital quantum number) for each peak, and the system-dependent Fermi energy shift[26]. While the double-peak feature observed in specific heat allows us to estimate the $g$-factor, without having to make any assumptions concerning the Landau index or Fermi energy shift. As a first approach, it is possible to estimate the electron and hole $g$-factors, implicitly involved in the DOS model, from the magnetic fields $(B_0^{\uparrow\downarrow})$ at which the spin Landau levels cross $E_F$. The crossing condition gives $g = 2(dE/dB)(B_0^\downarrow - B_0^\uparrow)/\mu_B(B_0^\downarrow + B_0^\uparrow)$. Using the values in Table 1,

we obtain $g = 2.0, 2.2$, and $2.0$ for the $1_e$, $2_e$ and $1_h$ Landau levels respectively. These values are close to the free electron $g$-factor due to the small spin-orbit coupling of the carbon atom[27], and in good agreement with electron-spin-resonance measurements in graphite[28–31].

It is clear that our simple DOS model provides a reasonable estimate of the $g$-factor. However, in most cases of quantum oscillations, the exact shape of the DOS is unknown, making it difficult to fit the data in order to extract the $g$-factor. Alternatively, this limitation can be overcome by using a coincidence method based on the magnetic field positions of the double peaks. The specific heat which depends on an integral involving the kernel term $-x^2 dF(x)/dx$ represents a spectroscopic tuning fork of width $4.8 k_B T$ which can be tuned at will to resonance. For example, the observed coincidence of the $B_1$ and $B_2$ features of the $1_h^{\pm}$ spin Landau levels at $T = 1.09$ K and $B = 3.75$ T (marked as orange arrow in Fig. 2b), corresponds to the condition where the spin-split Landau levels simultaneously cross the maxima in $-x^2 dF(x)/dx$, i.e. $g_h \mu_B B = 5.8 k_B T$ (here we use the apparent splitting in $C_{el}/T$ due to Landau level width - see Supplementary Note 10 for details) allowing us to extract the hole $g$-factor $g_h = 2.49$. Likewise, the extrapolated crossing of the $1_e^{\pm}$ spin Landau levels at $T = 2.05$ K and $B = 7.42$ T provides an estimate for the electron $g$-factor $g_e \mu_B B = 5.9 k_B T$ (apparent splitting in $C_{el}/T$) gives $g_e = 2.42$. These values compare well with the accepted value of the electron/hole $g$-factor $g_s = 2.50$ used to fit de Haas van-Alphen data using the SWM Hamiltonian in natural graphite[21]. Note, $g$-factors measured by electron spin resonance[28–31] are smaller ($g = 2.15$) as they measure the single particle spin gap, while transport techniques measure the exchanged-enhanced spin gap. We emphasis that extracting the $g$-factor using both DOS model and coincidence method does not require the knowledge of Landau index and Fermi energy shift, which is an advantage beyond other techniques (see more discussion in Supplementary Note 14).

## Double-peak structure in the Lifshitz transition

The origin of the double-peak structure reported here is not restricted to the Landau quantization, but also applies to any system where a femionic sharp DOS peak is tuned by the magnetic field. For example, a double-peak structure was observed in the vicinity of the Lifshitz transition for heavy fermion compounds $CeRu_2Si_2$[32] and UCoGe (Supplementary Note 12, 13). In graphite, $dE/dB$ of Landau levels extracted from the double-peak structure originates from the cyclotron/Zeeman energies corrected for the Fermi energy shift in the magnetic field. To understand what drives the double-peak structure near the Lifshitz transition, we compare the measured $dE/dB$ with the field dependence of the cyclotron/Zeeman energies in Table 2. Clearly, the $dE/dB$ values extracted from the double-peak structure in both UCoGe and $CeRu_2Si_2$ are too large to be explained by the Zeeman/cyclotron energy of the heavy quasiparticles ($dE/dB \gg \hbar e/m^*$, $g_j m_j \mu_B$). We conclude that the main contribution to the $dE/dB$ near the Lifshitz transition is the shift of the Fermi energy. For example, in the case of $CeRu_2Si_2$, the field-induced valence instability is expected to induce a large shift of the Fermi energy[33]. Therefore, the double-peak structure in $C_{el}/T$ can potentially be used to determine the Fermi energy shift in the vicinity of Lifshitz transition, a physical quantity that is not easy to access using other probes.

## Kernel term for different probes

It is interesting to consider what other thermodynamic and transport properties of a double-peak structure are to be expected within the free electron theory. The exact form is an integral involving a convolution of the density of states $D(E)$ and a kernel term $-x^n dF(x)/dx$ with $n = 0, 1, 2$ depending upon the probe considered (see Supplementary Note 6)[34,35]. Figure 5 shows the three different kernel terms and the corresponding techniques. Although the shape of experimental data for different probes can be influenced by the $D(E)$ or other factors (e.g. scattering time for transport probes), it is the kernel term that determines the shape of the experimental data to be single- or double-peak feature. The single peak feature predicted for conductance and magnetization is well known[18,36]. The predicted positive and negative peaks in thermopower have also been observed experimentally in graphite (see Supplementary Fig. 7)[8,37]. With the single- and double-peak features of MCE and specific heat reported in the present study, the currently only unverified probe is thermal transport. As seen in Fig. 5, as thermal conductance has the same kernel term as the specific heat, simple theory predicts a similar double-peak structure in thermal transport. To the best of our knowledge, a double peak feature has yet to be observed in thermal transport, so we expect our results should stimulate further research in this direction.

To conclude, we have shown that, as the quantum limit is approached in high-quality graphite, the electronic specific heat divided by temperature $C_{el}/T$ exhibits a double-peak structure when a single spin Landau level crosses the Fermi energy that vanishes as $T \to 0$. A simple DOS model, combined with the predictions of the SWM

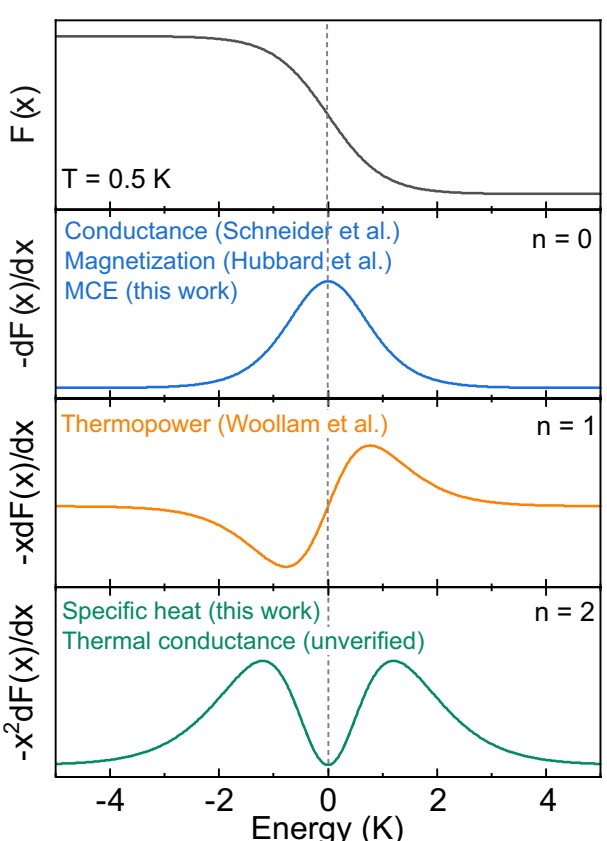

**Fig. 5 | Kernel terms for various thermodynamic and transport probes.** Fermi–Dirac distribution function $F(x)$ with $x = E/k_B T$, and the kernel terms $-x^n dF(x)/dx$ of the exact formula predicting the behavior of different thermodynamic and transport probes. All the curves were calculated at $T = 0.5$ K. The features of conductance, magnetization, and thermopower have been experimentally verified[8,18,36]. The presence of a double peak structure in thermal transport is currently a theoretical prediction.

**Table 2 | Summary of cyclotron, Zeeman coefficients, together with the experimentally determined energy shift $dE/dB$ from the double-peak structure, for UCoGe and $CeRu_2Si_2$[41–44]**

| compound | $B_c$ (T) | $\hbar e/m^*$ | $g_j m_j \mu_B$ | $dE/dB$ | Unit |
|---|---|---|---|---|---|
| UCoGe | 9.5 | 0.008 | 0.004 | 0.37 | meV/T |
| $CeRu_2Si_2$ | 7.7 | 0.078 | 0.109 | 1.05 | meV/T |

Hamiltonian, successfully reproduces the double-peak structure, which can be understood with the exact form of the free electron expression for $C_{el}/T$. The specific heat, which depends on an integral involving the kernel term $(-x^2 \mathrm{d}F(x)/\mathrm{d}x)$, represents a spectroscopic tuning fork of width $4.8 k_B T$ that can be tuned at will to resonance. Using a coincidence method, the double-peak structure provides a reliable estimate of the exchange-enhanced $g$-factor. Crucially, the double-peak structure is also observed in the specific heat of heavy-fermion compounds in the vicinity of the Lifshitz transition, potentially providing direct access to the Fermi energy shift at the Lifshiz transition.

## Methods
### Sample description
The measurements were performed on high-quality natural graphite samples. The graphite flakes have a typical length of $\simeq 0.1$ mm and a thickness of $\simeq 0.1$ mm. The weight of Sample#1–Sample#3 are 0.96 mg, 0.18 mg, 0.23 mg respectively.

### Experimental setup
AC-specific heat measurements were performed in a static magnetic field on natural graphite samples. During the experiment, the specimen was attached to the backside of a bare CERNOX resistive chip by a minute amount of Apiezon grease. The resistive chip was split into heater and thermometer part by artificially making a notch along the middle line of the chip. The heater part was used to generate a periodically modulated heating power $P_{ac}$ with a frequency of $2\omega$, which can be described as the following relation,

$$P_{ac} = \frac{R_H i_{ac}^2}{2\omega}, \tag{S4}$$

where $R_H$ is the resistance of heater part, $i_{ac}$ is a modulating current with a frequency of $\omega$. The induced oscillating temperature $T_{ac}$ of the sample was monitored by the thermometer part of the resistive chip. To do so, we applied a DC reading current $i_{DC}$ and monitored the induced AC voltage $V_{ac}$. Based on a precise calibration of the thermometer($R$-$T$ relation), $T_{ac}$ can be calculated from,

$$V_{ac}(2\omega) = \frac{\mathrm{d}R_T}{\mathrm{d}T} T_{ac}(2\omega) i_{DC}, \tag{S5}$$

Knowing $P_{ac}$ and $T_{ac}$, specific heat can be calculated by[38],

$$C = \frac{P_{ac} |\sin(\phi)|}{2\omega |T_{ac}|}, \tag{S6}$$

Here, $\phi$ stands for the phase shift between $P_{ac}$ and $T_{ac}$. By properly choosing the measurement frequency ($\omega$), $\phi$ is close to -90$^o$ ($|\sin(\phi)| \simeq 1$).

To measure the angle dependence of the specific heat in the magnetic field, a CERNOX resistive chip is mounted on a copper ring attached to an attocube rotator. On the back of the copper ring, a Hall probe allows us to measure the angle with the magnetic field. The misalignment between the sample and the Hall probe is estimated to be within ± 2 degrees.

MCE measurement was carried out in long pulsed fields with a duration of 1.2 s. The temperature of the natural graphite sample was read by the home-made RuO$_2$ thermometer, which was calibrated in temperature and magnetic field[39]. The temperature of the sample was monitored and recorded during the pulse field sweeps. For both measurements, the magnetic field was applied along the $c$-axis.

### SWM Hamiltonian
Graphite is a semi-metal with the carriers occupying a small region along the $H - K - H$ edge of the hexagonal Brillouin zone. The SWM Hamiltonian[15,16] with its seven tight binding parameters $\gamma_0, ..., \gamma_5, \Delta$ provides a remarkably accurate description of the band structure of graphite[17,18]. In a magnetic field, when trigonal warping is included ($\gamma_3 \neq 0$) levels with orbital quantum number $N$ couple to levels with orbital quantum number $N+3$ and the Hamiltonian has infinite order. Nevertheless, the infinite matrix can be truncated and numerically diagonalized, as the eigenvalues converge rapidly[40].

The values of SWM parameters that are used in this study are shown in Supplementary Table 1, taken from the SWM parameter set optimized to fit de Haas–van Alphen measurements in natural graphite[21]. They vary very little from the published values in other reports, e.g.[10,18,19].

## Data availability
All other data that suppot the findings of this study are available upon request to the corresponding author. Source data are provided with this paper.

## Code availability
The code for the SWM Hamiltonian calculation is available upon request to the corresponding author.

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

## Acknowledgements

The authors acknowledge the support of the LNCMI-CNRS, a member of the European Magnetic Field Laboratory (EMFL). This study has been partially supported through the EUR grant NanoX no. ANR-17-EURE-0009 in the framework of the "Programme des Investissements d'Avenir" and the Japan Society for the Promotion of Science (JSPS) KAKENHI Grants-In-Aid for Scientific Research (Nos. 22H00104, 20K14403), UTEC-UTokyo FSI Research Grant Program. B.F. is supported by JEIP-Collège de France. This work is also supported by the EU H2020 project: European Microkelvin Platform (EMP), grant agreement No. 824109. Z.P. and J.K. also acknowledge the support by the Slovak Research and Development Agency, under Grant No. APVV-20-0425 and by Slovak Scientific Grant Agency under Contract VEGA-0058/20.

## Author contributions

Z.Y. and Y.K. conceived the study. B.F. provided high-quality natural graphite samples and performed transport measurements. A.P., G.K., and D.A. grew high quality UCoGe single-crystalline sample. C.M., T.K., Z.P., and J.K. performed specific heat measurements. Y.K., T.S., and T.N. performed MCE measurements. Z.Y. and D.M. analyzed the data and performed the simulation. Z.Y., D.M., Y.K. C.M., T.K., B.F., D.C., and S.K. discussed and interpreted the results. Z.Y., D.M., and Y.K. prepared the manuscript, with input from all other co-authors.

## Competing interests

The authors declare no competing interests.
