## [Peer Review File · Nature Communications]

REVIEWER COMMENTS

Reviewer #1 (Remarks to the Author):

Zhuo Yang and colleagues report on their study of quantum oscillations (QOs) of electronic specific heat (C_{el}) in natural graphite. Normally, QOs measurements are conducted using resistivity, magnetization, thermoelectricity, and other methods. However, detecting the QOs amplitude of C_{el} is notoriously difficult due to its typically small signal, as previously noted by Paul F. Sullivan and G. Seidel (Phys. Rev. 173, 679 (1968), *物理学报* Acta Phys. Sin. 63, 240502(2014)). As a result, quantum oscillation of C_{el} has been less frequently reported in the scientific literature. The authors of the current study discovered that high-quality QOs of C_{el} display a double-peaked structure that disappears as the temperature approaches absolute zero. They were able to explain this observation through a theory based on free electrons. Additionally, the authors identified a 4.8kBT resonance in specific heat spectroscopy and were able to resolve the Lande factor g of graphite. The authors further checked this finding in heavy-fermion compounds.

I think the manuscript is ready for publication, but I do have a few points to raise:

1. In this article (*物理学报* Acta Phys. Sin. 63, 240502(2014)), the theoretical prediction of the double-peak structure for specific heat oscillations is discussed, and two models of oscillations are presented. Although there may be some differences in the systems studied, there are also some similarities that may warrant further discussion in the authors' manuscript.
2. Equation S3 implies that the differential of entropy over temperature is the electronic specific heat over temperature. It would be interesting if the authors could measure two entropy curves of field-dependent entropy at two adjacent temperatures ($1/T$) (similar to the curve shown in Figure 1a). Then, the difference in field between the two curves could be compared with Figure 1b.
3. Following on from the previous point, upon careful inspection of the curves, it appears that the $1h+$ and $1e+$ peaks in the curve of $1/T$ in the Fig.1a are at the same position as those of C_{el}/T in the Fig.1a. However, the other peaks do not seem to be at the same field positions. According to the authors' argument, the peaks of $1/T$ are located in the middle of the double peaks of C_{el}/T with the same Landau index. Since the origin of the double peaks is the extension of Landau crossing from thermal effects (Fig. 3 and Fig. S5), this point may require further clarification.

4. To enable better comparison, I suggest that the authors measure resistivity or magnetization and include them in Figure 1a.

5. In principle, QOs from other methods carry the same information regarding the g -factor and other parameters (e.g., thermal effects) as specific heat. By performing appropriate calculations and fitting the data, it should be possible to resolve the g -factor. For example, the Γ parameter in the specific heat equation can be used to determine dE/dB . If QOs from other methods are fitted while considering thermal effects, the dE/dB should also be obtained. The thermal broadening effect, which is responsible for the broadening of QO peaks in other methods, should also be at the order of 4.8 kBT. The authors should elaborate more on the implications for other methods, as discussed on page 13 of the supplementary information.

6. Regarding Fig.S5, the $SdH/dHvA$ be marked clearly as this?

7. The explanation for the double peaks observed in heavy-fermion systems may be problematic. In a previous study (J. Magn. Magn. Mater. 177, 271 (1998)), the authors argued that the two peaks were due to the Zeeman shift of the band, as shown in Figure 5 of reference 31. This differs from the thermal effect of specific heat suggested by the current authors. Moreover, the double peaks were also observed in thermal expansion measurements (J. Magn. Magn. Mater. 177, 271 (1998)), C. Paulsen et al., J. Low Temp. Phys. 81, 317 (1990)). Therefore, the double peaks observed in heavy-fermion materials may have a different source. Additionally, the many single peaks for Landau levels observed in Figure S1b suggest that the double peak may not be necessary.

Please also see the attached file.

Reviewer #2 (Remarks to the Author):

A quantitative analysis of clear quantum oscillations in the electronic specific heat is reported using high-quality graphite. A double-peak structure appears in specific heat due to the crossing of a spin-split Landau level and the Fermi energy. The double-peak structure, as well as its temperature dependence, is well explained by the free electron theory using SWM tight-binding model. Using this model, the Lande g -factor of graphite is accurately determined in agreement with the previous

reports with different measurement techniques. Furthermore, the authors point out that a similar analysis can be applied to other systems in the vicinity of the Lifshitz transition in heavy-fermion compounds.

The data quality is very high and a detailed quantitative analysis is performed in the present study. However, I'm not convinced how this technique provides us with new information on quantum materials and cannot see any broader impact that can open a new direction in research to justify publication in Nature Communications.

Comment 1) It is mentioned that the estimation of the g-factor is difficult in graphite on lines 224 and 225. However, clear Zeeman splitting is observed even in MCE measurements as shown in Fig. 1(a). If you can observe Zeeman splitting in MCE, then what is the advantage of extraction of g-factor using electronic specific heat?

Comment 2) The heat capacity results on heavy fermion compounds are interesting. The large value of dE/dB likely originates from the large shift of Fermi energy at the Lifshitz transition. However, can you perform more quantitative analysis, such as estimating the Fermi energy shift?

Comment 3) Why are there no data points for peaks B1 and B2 of $1e^+$ and $1e^-$ at 0.1 and 0.3 K in Fig. 2(b)?

Comment 4) Can you compare the parameter Γ with the quantum scattering time ($\tau_{_Q}$) extracted from the Dingle plot of Shubnikov-de Haas oscillations from previous studies? The energy broadening can be estimated as the inverse of $\tau_{_Q}$, so this comparison should also confirm the verification of your analysis.

Reviewer #3 (Remarks to the Author):

The authors reported a double-peak effect in the specific heat oscillations of natural graphite single crystal sample, which is very interesting. I have a few questions/comments to the experiments: 1) are these results repeatable in different samples? 2) is there any extrinsic effect, e.g., twin crystal or sample mounting fault? 3) is there any other extrinsic effect from the measurements, e.g., the temperature-dependent double peak effect shown in Fig. 10d of Ref. 33? The measurements are conducted with magnetic field along the c axis of sample. The intrinsic effect in graphite follows

quasi-2D rule in magnetic fields, therefore, the rotation angle measurements are required to support the claims in this manuscript.

Further, it is very hard to understand the single spin Landau level splitting effect, which is the cornerstone of the fitting model. The renormalization of Fermi level to keep charge neutrality leads to broadening peak accordingly, not splitting it into double peaks. In not the case in CeRu₂Si₂, it is very clear that the splitting occurs between subbands with different spins. In Fig. 2a, the peak intensities also show anomalous temperature dependency, e.g., the B1 peak is stronger below 0.5 K than B2, however much weaker >1 K, why? The authors mentioned anomaly in thermal transport a few times in their manuscript, however they didn't provide any evidence for that. As an unverified hypothesis, it seems too early to discuss this point in the present manuscript.

Therefore, I cannot accept its publication unless the authors provide more convincing evidence to support the claims.

Unveiling hidden double-peak structure of quantum oscillations in the specific heat

Zhuo Yang, Benoît Fauqué, Toshihiro Nomura, Takashi Shitaokoshi, Sunghoon Kim, Debanjan Chowdhury, Zuzana Pribulová, Jozef Kačmarčík, Alexandre Pourret, Georg Knebel, Dai Aoki, Thierry Klein, Duncan K. Maude, Christophe Marcenat and Yoshimitsu Kohama

REVIEWER COMMENTS

Reviewer #1 (Remarks to the Author):

Referee: Zhuo Yang and colleagues report on their study of quantum oscillations (QOs) of electronic specific heat (C_{el}) in natural graphite. Normally, QOs measurements are conducted using resistivity, magnetization, thermoelectricity, and other methods. However, detecting the QOs amplitude of C_{el} is notoriously difficult due to its typically small signal, as previously noted by Paul F. Sullivan and G. Seidel (*Phys. Rev.* **173**, 679 (1968), *物理学报 Acta Phys. Sin.* **63**, 240502(2014)). As a result, quantum oscillation of C_{el} has been less frequently reported in the scientific literature. The authors of the current study discovered that high-quality QOs of C_{el} display a double-peaked structure that disappears as the temperature approaches absolute zero. They were able to explain this observation through a theory based on free electrons. Additionally, the authors identified a $4.8k_B T$ resonance in specific heat spectroscopy and were able to resolve the Lande g -factor of graphite. The authors further checked this finding in heavy-fermion compounds.

I think the manuscript is ready for publication, but I do have a few points to raise:

Authors: We thank the referee for the positive appraisal of our work and suggestions for changes, which we address below.

Referee: Comment 1) In this article (*物理学报 Acta Phys. Sin.* **63**, 240502 (2014)), the theoretical prediction of the double-peak structure for specific heat oscillations is discussed, and two models of oscillations are presented. Although there may be some differences in the systems studied, there are also some similarities that may warrant further discussion in the authors' manuscript.

Authors: We thank the referee for mentioning this interesting article (*物理学报 Acta Phys. Sin.* **63**, 240502 (2014)) that we were unaware of. We have now added a discussion (manuscript line 158) and cited this article.

Changes:

I. Manuscript

Page 8, line 158: “The double-peak structure in quantum oscillations was also predicted in earlier theoretical calculations using an explicit expression for the specific heat [23]”

Page 21, line 432: we added the reference “[23] Z.-Q. Shao, J.-W. Chen, Y.-Q. Li, and X.-Y. Pan, Thermodynamical properties of a three-dimensional free electron gas confined in a one-dimensional harmonical potential, *Acta Phys. Sin.* **63**, 240502 (2014)”

Referee: Comment 2) Equation S3 implies that the differential of entropy over temperature is the electronic specific heat over temperature. It would be interesting if the authors could measure two entropy curves of field-dependent entropy at two adjacent temperatures ($1/T$) (similar to the curve shown in Figure 1a). Then, the difference in field between the two curves could be compared with Figure 1b.

Authors: We thank the referee for this very interesting suggestion. We fully agree with the referee and have added a discussion in **SI Sec. VII**.

In the revised SI, **Fig S6(c)** shows the calculated temperature derivative of the entropy $S(2K)-S(1K)/1K$ as a function of the magnetic field, which display a double-peak structure as anticipated by the referee.

Unfortunately, with our current set of data on MCE, we are not able to resolve it experimentally. We leave this for future more accurate measurements focusing on the study of this double-peak structure in the temperature differential MCE.

Changes:

II. Supplementary information

Page 13, line 177: “However, the differential entropy $dS_{el}/dT \propto C_{el}/T$ is expected to show the double peak structure...”

Page 14, Fig. S6: we added panel (c) and revised the caption.

Referee: Comment 3) Following on from the previous point, upon careful inspection of the curves, it appears that the 1_h^+ and 1_e^+ peaks in the curve of $1/T$ in the Fig.1a are at the same position as those of C_{el}/T in the Fig.1a. However, the other peaks do not seem to be at the same field positions. According to the authors' argument, the peaks of $1/T$ are located in the middle of the double peaks of C_{el}/T with the same Landau index. Since the origin of the double peaks is the extension of Landau crossing from thermal effects (Fig. 3 and Fig. S5), this point may require further clarification.

Authors: We thank the referee for pointing out this inaccurate statement.

The deviation is mainly attributed to two reasons:

1. The peaks in $1/T$ are located at the centre of the double peak structure in C/T provided the DOS crossing E_F is symmetric, e.g., cusp-like DOS peak in Lifshitz transition. However, in the case of Landau levels for 3D materials, the DOS display a succession of van Hove singularities that are highly asymmetric, which induces certain deviation between the peak position in $1/T$ and centre of double-peak structure in C/T .

2. Both MCE and C_{el}/T data in Fig. 1 were obtained by subtracting a Schottky contribution. This process can induce slight discrepancies between different techniques. Note that the deviation between MCE and C_{el}/T is less than 0.1 T.

We have revised the non-rigorous statement in the manuscript line 138 and added a section (SI Sec. XI) to discuss the effect of the asymmetry of the DOS on the peak position in $1/T$ and C_{el}/T .

Deletion:

Page 7, line 138: “center of”

Changes:

I. Manuscript

Page 12, line 230: “In addition, the asymmetric DOS peak also induces deviation between the peak position in $1/T$ and the center of double-peak structure (Supplementary Sec. XI).”

II. Supplementary information

Page 20, line 236: we added section “XI. Peak position in MCE and specific heat for symmetric and asymmetric DOS”

Referee: Comment 4) To enable better comparison, I suggest that the authors measure resistivity or magnetization and include them in Figure 1a.

Authors: We thank the referee for the suggestion; we have additionally measured magnetoresistance on natural graphite and included it in Fig. 1b.

Changes:

I. Manuscript

Page 5, Fig. 1: We added magnetoresistance data as panel (b)

Page 4, line 90: “For better comparison, Fig. 1(b) shows background removed magnetoresistance ΔR_{xx} on natural graphite at 0.5 K”

Referee: Comment 5) In principle, QOs from other methods carry the same information regarding the g -factor and other parameters (e.g., thermal effects) as specific heat. By performing appropriate calculations and fitting the data, it should be possible to resolve the g -factor. For example, the Γ parameter in the specific heat equation can be used to determine dE/dB . If QOs from other methods are fitted while considering thermal effects, the dE/dB should also be obtained. The thermal broadening effect, which is responsible for the broadening of QO peaks in other methods, should also be at the order of $4.8 k_B T$. The authors should elaborate more on the implications for other methods, as discussed on page 13 of the supplementary information.

Authors: We fully agree that an appropriate fitting to the QOs in other methods allows us to obtain the g -factor and dE/dB . However, fitting to the QOs requires a detailed knowledge of the DOS shape, which is generally unknown. The advantage of techniques showing double-peak structure (specific heat, thermal transport) allows us to extract g -

factor and dE/dB based mainly on the peak positions, which greatly simplifies the analysis.

It is interesting to note that thermopower exhibit a maximum and a minimum for a single DOS peak passing the Fermi level. We predicted that the splitting of maximum/minimum in thermopower is $3.09 k_B T$. Knowing this, the dE/dB and g -factor can also be extracted from QO of thermopower in a similar manner like in specific heat.

In the revised manuscript/SI, we discuss extensively the various thermodynamic and transport probes based on the kernel term $-x^n dF/dx$ in manuscript subsection “Kernel term for different probes” and SI Sec. VI – IX.

Changes:

I. Manuscript

Page 14, line 292: We added section “Kernel term for different probes”

II. Supplementary information

Page 12, line 141: We added section “VI. Exact form for various thermodynamic probes”

Page 16, line 195: We added section “IX. Predictions for thermopower”

Referee: Comment 6) Regarding Fig.S5, the SdH/dHvA be marked clearly as this?

Authors: We apologise for the misleading schematic figure which was indeed badly labelled. It was over ambitious to attempt to produce a satisfactory schematic LL fan chart for the different probes without being misleading. We prefer to remove the figure in question.

Instead, we discuss different probes based on kernel term $-x^n dF/dx$ as mentioned in previous question. We believe this way provides a more profound understanding to the differences between various techniques.

Deletion:

II. Supplementary information

We deleted the whole section “VII. On the way to the Landau level slope (dE/dB)”

Referee: Comment 7) The explanation for the double peaks observed in heavy-fermion systems may be problematic. In a previous study (*J. Magn. Magn. Mater.* **177**, 271 (1998)), the authors argued that the two peaks were due to the Zeeman shift of the band, as shown in Figure 5 of reference 31. This differs from the thermal effect of specific heat suggested by the current authors. Moreover, the double peaks were also observed in thermal expansion measurements (*J. Magn. Magn. Mater.* **177**, 271 (1998)), C. Paulsen et al., *J. Low Temp. Phys.* **81**, 317 (1990)). Therefore, the double peaks observed in heavy-fermion materials may have a different source.

Authors: We agree that the double-peak structure in the Lifshitz transition of CeRu_2Si_2 is induced by the Zeeman shift. However, we believe that it is only caused by ONE of the spin-split subbands, in other words, there is only one sharp DOS peak crossing the Fermi energy in the Lifshitz transition. As seen in SI Sec. XIII, our single DOS peak model

provides excellent fitting to the C/T data of CeRu_2Si_2 . Another direct evidence is that the MCE of CeRu_2Si_2 only shows a single peak feature (Fig. 4 in *J. Magn. Magn. Mater.* **177**, 271 (1998)). This is exactly the same feature as we observed in graphite: single- and double-peak structure in MCE and specific heat are the thermodynamic evidence for a single DOS peak crossing the E_F .

The scenario that a single spin-split subband passes through E_F in CeRu_2Si_2 has also been demonstrated by magneto-transport (*Phys. Rev. Lett.* **96**, 026401 (2006)), thermopower (*Phys. Rev. B* **85**, 035127 (2012)).

Regarding the thermal expansion measurement, in the previous study (C. Paulsen et al., *J. Low Temp. Phys.* **81**, 317 (1990)), there is no field sweep of thermal expansion data, but only temperature sweep data. Therefore, we could not make a direct comparison between the thermal expansion and specific heat. However, we agree that the field sweep of thermal expansion measurement may also show a double-peak structure, because thermal expansion is closely related to the specific heat.

Changes:

II. Supplementary information

Page 22, line 252: We added section “XIII. Fitting of C/T near the Lifshitz transition in CeRu_2Si_2 ”

Referee: Comment 8) Additionally, the many single peaks for Landau levels observed in Figure S1b suggest that the double peak may not be necessary.

Authors: We thank the referee for raising up this interesting point. There is a general belief in the literature that the Landau level in C/T is a single peak feature, while our results demonstrated it is indeed a double-peak feature. Without the knowledge of double-peak structure, it is easy to make mistakes, e.g., the frequency of Fermi surface will be two times higher than the real case, when we analyse the quantum oscillation in the double-peak structure region. Therefore, we believe it is important to clarify this point by presenting both cases.

Changes:

II. Supplementary information

Page 3, line 50: “which has been widely used in the literature [2–5]. Together with the double-peak structure...”

Reviewer #2 (Remarks to the Author):

Referee: A quantitative analysis of clear quantum oscillations in the electronic specific heat is reported using high-quality graphite. A double-peak structure appears in specific heat due to the crossing of a spin-split Landau level and the Fermi energy. The double-peak structure, as well as its temperature dependence, is well explained by the free electron theory using SWM tight-binding model. Using this model, the Lande g-factor of graphite is accurately determined in agreement with the previous reports with different measurement techniques. Furthermore, the authors point out that a similar analysis can be applied to other systems in the vicinity of the Lifshitz transition in heavy-fermion compounds.

The data quality is very high and a detailed quantitative analysis is performed in the present study. However, I'm not convinced how this technique provides us with new information on quantum materials and cannot see any broader impact that can open a new direction in research to justify publication in Nature Communications.

Authors:

We highly thank the referee for constructive suggestions. However, we respectfully disagree with the last statement of the referee. Our result will impact the understanding of the ground state of a large variety of quantum materials for the following reasons:

1. Correction to general belief

There is a general belief in the literature that double peaks in $C(B)/T$ corresponds to two peaks in the DOS crossing the Fermi energy E_F . We demonstrate that it is not the case in two completely different situations : the quantum oscillations of graphite and the Lifshitz transition of heavy Fermion $CeRu_2Si_2$ and $UCoGe$. Our results provide thus a major change in the understanding of one of the most fundamental thermodynamic quantity.

2. Probing the nature of DOS peak

The double-peak structure in $C(B)/T$ reported here provide also a new tool to quantify the nature of the DOS. It provides information on the shape (symmetric/asymmetric), linewidth (Γ), shift (dE/dB) and allow to access to microscopic parameters such as the g-factor of the charge carrier. All these information are notoriously difficult to quantify otherwise but are of fundamental interest to understand the ground state of quantum materials.

3. Impact to correlated electron system - logarithmic divergence in $C(T)/T$

For the temperature dependence of $C(T)/T$, the single DOS singularity crossing E_F induce "logarithmic divergence in $C(T)/T$ " (Fig. R1(c) and Fig. S11(b)), namely $C/T \propto \log(T_0/T)$. The logarithmic divergence is thought to be a signature of non-Fermi-liquid behaviour (*Phys. Rev. Lett.* **93**, 096402 (2004)). For scientists working on correlated electron systems, it should be important to notice that the crossing of the single DOS singularity at E_F can induce the similar behaviour. Thus, our paper indicates that the logarithmic divergence in $C(T)/T$ is not only due to an existence of a canonical quantum critical point,

but can also be attributed to a formation of fermionic DOS peak (e.g., Landau quantization).

4. Other potential systems

This double peak structure detected here is not limited to QOs and Lifshitz transition, but to any systems where the DOS is characterised by a single peak tuned by the magnetic field. Here, we listed a few systems that can be potentially studied by our technique.

Sr₃Ru₂O₇ (*Science* **294**, 329-332 (2001))

YbRh₂Si₂ (*Phys. Rev. Lett.* **110**, 256403 (2013))

Ca_{1.8}Sr_{0.2}RuO₄ (*Commun. Phys.*, **4**(1), 1 (2021))

CeTiGe (*Phys. Rev. B*, **85**, 060401, (2012))

Let's finally point that such tools could be also apply to magnetic insulator displaying fermionic excitations (P. G. LaBarre et al. arXiv:2111.03758 (2021)).

Being far from a curiosity the double-peak structure can be used as a fine probed of the DOS in many different situations and in many other quantum materials. There is thus no doubt that our result will bring new information on their remarkable ground states.

Fig. R1 (a) Constructed symmetric DOS singularity that crosses E_F at $B_0 = 7.7$ T. (b) Calculated C/T versus B at indicated temperature. (c) Calculated C/T versus T . The temperature-sweep of C/T at critical field $B = 7.7$ T shows logarithmic divergence ($C/T \propto \log(T_0/T)$).

Referee: Comment 1) It is mentioned that the estimation of the g -factor is difficult in graphite on lines 224 and 225. However, clear Zeeman splitting is observed even in MCE measurements as shown in Fig. 1(a). If you can observe Zeeman splitting in MCE, then what is the advantage of extraction of g -factor using electronic specific heat?

Authors: We thank the referee for the question. In usual QOs techniques such as SdHs, dHvA, MCE, to extract the g -factor, one has to know the Landau index (orbital quantum number) for each peak, and include the field dependence of the Fermi energy for low carrier metals. On the other hand, the double-peak structure observed in C_{el}/T allows us to estimate the g -factor, without any assumptions concerning the Landau index or Fermi energy shift.

To clarify this key result, we have emphasized this point in the manuscript, and added a new section in the SI (SI Sec. XIV) that present the advantage of this new tool in comparison to usual ones.

We also agree with the referee that the g -factor in graphite is a well-known quantity. However, the aim of our manuscript is, using graphite as an example, to demonstrate a novel tool that allows to estimate the g -factor, which can be apply to many other materials.

Changes:

I. Manuscript

Page 12, line 243: “In general, to extract the g -factor using techniques such as SdHs, dHvA, MCE etc, one has...”

Page 13, line 254: “It is clear that our simple DOS model provides a reasonable estimate...”

Page 13, line 271: “We emphasis that extracting the g -factor using both DOS model...”

II. Supplementary information

Page 24, line 281: We added section “XIV. Advantage of extracting effective g -factor from double-peak structure in C_{el}/T ”

Referee: Comment 2) The heat capacity results on heavy fermion compounds are interesting. The large value of dE/dB likely originates from the large shift of Fermi energy at the Lifshitz transition. However, can you perform more quantitative analysis, such as estimating the Fermi energy shift?

Authors: Yes, we can estimate the Fermi energy shift of the Lifshitz transition. They are $S_F = 0.362$ meV/T for UCoGe and $S_F = 0.902$ meV/T for CeRu₂Si₂.

To estimate the shift of the Fermi energy, we need to subtract the Zeeman and cyclotron contributions from the shift of the DOS peak (dE/dB), namely,

$$S_F = \frac{dE}{dB} - \frac{\hbar e^*}{2m^*} - g_j m_j \mu_B$$

The values of dE/dB , $\hbar e/2m^*$ and $g_j m_j \mu_B$ are 0.37, 0.004, 0.004 meV/T for UCoGe, and 1.05, 0.039, 0.109 meV/T for CeRu₂Si₂ (see also the manuscript Table. II). As suspected by the referee, the major contribution to dE/dB at the Lifshitz transition is the

shift of the Fermi energy. To our best knowledge, this is the first estimation of the Fermi energy shift at the Lifshitz transition using specific heat.

In addition, we show the fitting to the C/T of CeRu_2Si_2 in revised **SI Sec. XIII**. This fitting result suggest that the DOS peak at the Lifshitz transition of CeRu_2Si_2 is cusp-like shape with $\Gamma = 0.085$ meV. The cusp-like DOS in CeRu_2Si_2 is in consistent with early theoretical prediction (*J. Phys. Soc. Japan* **75**(3), 033704, (2006)).

It is interesting to note that the temperature-sweep of C/T for the double-peak structure shows the logarithmic divergence ($C/T \propto \log(T_0/T)$) at the critical field (**Fig. S11 (b)**). This suggests that the logarithmic divergence in $C(T)/T$ is not only due to an existence of a canonical quantum critical point, but can also be attributed to a formation of fermionic DOS peak (e.g., Landau quantization).

Referee: Comment 3) Why are there no data points for peaks B_1 and B_2 of 1_e^+ and 1_e^- at 0.1 and 0.3 K in Fig. 2(b)?

Authors: The ultra-low temperature C_{el}/T data at 0.1 K and 0.3 K were measured using a 5T superconducting magnet equipped with a dilution refrigerator. Unfortunately, the 1_e^+ and 1_e^- features are beyond the field range of our magnet.

Referee: Comment 4) Can you compare the parameter Γ with the quantum scattering time (τ_Q) extracted from the Dingle plot of Shubnikov-de Haas oscillations from previous studies? The energy broadening can be estimated as the inverse of τ_Q , so this comparison should also confirm the verification of your analysis.

Authors: We thank the referee for the important suggestion. The Γ extracted from our DOS model is in the range of 0.16 - 0.21 meV. It is in a good agreement with the quantum scattering time (τ_q) estimated from the onset of SdH oscillations at mK temperature. For natural graphite, the quantum scattering time τ_q is 3.3 – 4.6 ps, which gives the energy broadening ($\Gamma = \hbar/\tau_q$) of 0.14 - 0.2 meV.

We have added a section in SI (**SI. Sec. V**) that discuss in further detail on this comparison.

Changes:

I. Manuscript

Page 10, line 189: “Note that the FWHM Γ , obtained here by fitting C_{el}/T versus B , is very close to the Landau level broadening $\Gamma_q = \hbar/\tau_q$ determined from....”

II. Supplementary information

Page 11, line 130: We added section “V. Landau level broadening Γ_q and quantum lifetime τ_q in graphite”

Reviewer #3 (Remarks to the Author):

Referee: The authors reported a double-peak effect in the specific heat oscillations of natural graphite single crystal sample, which is very interesting. I have a few questions/comments to the experiments:

Authors: We thank the referee for the positive appraisal of our work, and suggestions for changes, which we address below.

Referee: Comment 1) are these results repeatable in different samples?

Authors: Yes, these results are repeatable. We performed additional measurements and observed double-peak structures in 3 different natural graphite samples, as shown in SI Sec. III. A.

Changes:

I. Manuscript

Page 4, line 101: “To verify that the double-peak structure in C_{el}/T is an intrinsic effect, we....”

II. Supplementary information

Page 5, line 70: We added subsection “III. A. Reproducibility of double-peak structure for different samples”

Referee: Comment 2) is there any extrinsic effect, e.g., twin crystal or sample mounting fault?

Authors: We thank the referee for this question. Extrinsic effect from the sample can affect the amplitude, or even smear out the double-peak structure. In the presence of high level of disorder, the energy broadening of Landau level is much larger than $4.8 k_B T$. Then the double-peak structure will be absent. However, we are not in this situation, because the line width of the Landau levels in natural graphite sample ($\Gamma = 0.16-0.21$ meV) is narrow enough to allow the observation of the double peak structure ($4.8 k_B T = 0.41$ meV at $T = 1$ K).

Referee: Comment 3) is there any other extrinsic effect from the measurements, e.g., the temperature-dependent double peak effect shown in Fig. 10d of Ref. 33?

Authors: We thank the referee for this interesting question. As explained in Ref. 33, the extrinsic effect in the specific heat measurement will lead to different shape of C_{el}/T curves between up- and down-sweep of the field owing to the opposite sign of $(\partial S/\partial B)_T (dB/dt)$, where $(\partial S/\partial B)_T$ and (dB/dt) are field dependence of isothermal entropy and field sweep rate, respectively. We now show in SI Section III. C the up- and down-sweep of field dependent C_{el}/T on natural graphite, they are identical. For this reason, we are confident that the double-peak structures do not result from an extrinsic effect from the measurement.

We also note that there are indeed three anomalies in $\text{Sr}_3\text{Cr}_2\text{O}_8$ resulting from Schottky anomaly, extrinsic effect of MCE and long-range order phase transition in Fig. 10d of Ref. 33, which are shown in the following figure panel (a). They are different from the double-peak structure reported in our work, because the characteristic feature for the double-peak structure is that the field positions of the two peaks are linearly shift with temperature. While the three anomalies in Ref. 33 are not consistent with the T -linear feature as discussed below:

1. **Long range order transition:** the field position of long-range order transition should follow $T^{3/2} - T^2$ dependence (*Phys. Rev. Lett.* **103**, 207203 (2009) & *J. Phys. Soc. Jpn.* **70**, 939 (2001)), but not T linear dependence, as seen in panel (b).
2. **Extrinsic effect of MCE:** the extrinsic effect appears at the magnetic field showing magnetocaloric effect, which is temperature independent (*Phys. Rev. B* **77**, 214441 (2008)).
3. **Schottky anomaly:** the peak position of Schottky anomaly is indeed T -linear dependence since the size of temperature shift is proportional to the Zeeman energy. However, it only provides a broad single peak.

To clarify the applicability of our model, we emphasis the T -linear dependence of the magnetic field position in double-peak structure in manuscript line 118.

Figure R2 (a) Field sweep specific heat of $\text{Sr}_3\text{Cr}_2\text{O}_8$ from ref. 33. The data was taken in the down-sweep of the magnetic field pulse. (b) Magnetic field positions of anomalies in ref. 33 as a function of temperature. The dashed lines are guide to the eyes to show the deviation of the linear temperature dependence.

Changes:

I. Manuscript

Page 4, line 101: "To verify that the double-peak structure in C_{el}/T is an intrinsic effect, we...."

Page 6, line 118: "The T -linear dependence of the peak positions B_1 , B_2 is a characteristic feature for the double-peak structure presented in this study."

II. Supplementary information

Page 7, line 87: We added subsection “III. C. Reproducibility between up and down magnetic field sweeps”

Referee: Comment 4) The measurements are conducted with magnetic field along the c axis of sample. The intrinsic effect in graphite follows quasi-2D rule in magnetic fields, therefore, the rotation angle measurements are required to support the claims in this manuscript.

Authors: To address the referee’s comments, we additionally measured angle-dependent C_{el}/T of the double-peak structure of 1_e^+ level for natural graphite (sample#2) at $T = 0.5$ K. The angle-dependent field position of B_1 and B_2 peaks for 1_e^+ levels follow $1/\cos(\theta)$ dependence up to 60 degrees - a signature for the quasi-2D behaviour of graphite.

We present this set of data in SI Sec. III. B.

Changes:

I. Manuscript

Page 4, line 101: “To verify that the double-peak structure in C_{el}/T is an intrinsic effect, we...”

II. Supplementary information

Page 6, line 76: We added subsection “III. B. Angle-dependence of double-peak structure”

Referee: Comment 5) Further, it is very hard to understand the single spin Landau level splitting effect, which is the cornerstone of the fitting model. The renormalization of Fermi level to keep charge neutrality leads to broadening peak accordingly, not splitting it into double peaks. In not the case in CeRu_2Si_2 , it is very clear that the splitting occurs between subbands with different spins.

Authors: We agree with the referee that the observation of a double-peak structure in C_{el}/T is surprising in presence of the single peak in DOS. This unanticipated striking result is the consequence of the effect of the kernel term $-x^n dF/dx$ with $n=2$, which appears as a double structure, as shown in Fig.3(a). The DOS peak and the renormalization of Fermi level are not responsible for the origin of double-peak structure, they only influence the shape or the splitting of the double peaks.

Note that the kernel terms are different for different probes. The double-peak structure is absent for the probes which have the kernel term $-x^n dF/dx$ with $n=0$ (e.g., conductance, MCE) and $n=1$ (thermopower). We provide in manuscript subsection “Kernel term for different probes” and SI Sec. VI a detail discussion on the effect of the kernel terms in various thermal dynamic and transport probes.

We also note that the double-peak structure is expected to merge into a single peak as the temperature approach 0 K, because the splitting ($4.8 k_B T$) of kernel function is smaller than the width of the DOS peak. We clarified this point in manuscript line 233.

Let's finally comment on the case of CeRu_2Si_2 . We do agree that the DOS is splitted in two levels due to the Zeeman shift of the subband. However, we believe it is only induced by ONE of the spin-split subbands. As seen in SI Sec. XIII, our peak model provides excellent fitting to the C/T data of CeRu_2Si_2 . Also, we note that MCE of CeRu_2Si_2 shows single-peak feature (Fig. 4b in *J. Magn. Magn. Mater.* **177**, 271 (1998)), which is consistent with the picture for the single DOS peak passing E_F . This scenario is in agreement with the picture from early transport (Fig. 3 in *Phys. Rev. Lett.* **96**, 026401 (2006)) and thermopower measurement (*Phys. Rev. B* **85**, 035127 (2012)).

Changes:

I. Manuscript

Page 12, line 233: "As $T \rightarrow 0$, we expect the double-peak structure to merge into a single peak..."

Page 14, line 292: We added section "Kernel term for different probes"

II. Supplementary information

Page 12, line 140: We added section "VI. Exact form for various thermodynamic probes"

Page 22, line 252: We added section "XIII. Fitting of C/T near the Lifshitz transition in CeRu_2Si_2 "

Referee: Comment 6) In Fig. 2a, the peak intensities also show anomalous temperature dependency, e.g., the B_1 peak is stronger below 0.5 K than B_2 , however much weaker >1 K, why?

Authors: The apparently anomalous temperature dependence arises simply from overlapping contributions of the spin up/down DOS "singularities", namely, B_2 peak of 1_{h^+} and B_1 peak of 1_{h^-} merge together at $T > 1$ K, as seen in the following Fig. R3(c).

This temperature-dependence is almost exactly reproduced by the theory, as seen from the following comparison between DOS model calculation (Fig. R3(b)) and experiment data (Fig. R3(c)).

Figure R3 (a) Constructed Landau level DOS that used to calculate the C_{el}/T . (b) Calculated C_{el}/T of 1_{h^+} and 1_{h^-} Landau levels at different temperatures. (c) Experimental data of C_{el}/T of 1_{h^+} and 1_{h^-} Landau level at different temperatures for a comparison. Note, this set of data is the same data as in Fig. 2(a) in the manuscript.

Referee: Comment 7) The authors mentioned anomaly in thermal transport a few times in their manuscript, however they didn't provide any evidence for that. As an unverified hypothesis, it seems too early to discuss this point in the present manuscript.

Authors: We thank the referee for spotting this unclear point. As C/T and thermal transport have exactly the same kernel term ($-x^2 dF/dx$), the double-peak feature should be observable in thermal transport. However, to the best of our knowledge, a double-peak structure has never been reported experimentally in thermal transport. Therefore, this point is included in the manuscript as a theoretical prediction.

We now make this point clear in the caption of Fig. 5 “The presence of a double peak structure in thermal transport is currently a theoretical prediction.” and give the exact kernel terms for various probes in **SI Sec. VI**.

Deletion:

line 163: “It is worth noting that $-x^2 dF(x)/dx$ is also involved in the exact form of the thermal conductance [39], suggesting a similar double-peak structure should be observable in thermal transport measurements”

Changes:

I. Manuscript

Page 14, line 292: We added section “Kernel term for different probes”

II. Supplementary information

Page 12, line 140: We added section “VI. Exact form for various thermodynamic probes”

Referee: Comment 8) Therefore, I cannot accept its publication unless the authors provide more convincing evidence to support the claims.

Authors: We thanks again to the referee for the constructive suggestions. In summary, we have introduced the following changes to the manuscript.

1. We performed additional measurements and present data showing the double-peak structure in three natural graphite samples, indicating a good reproducibility.
2. We additionally present angle-dependence of the double-peak structure. The magnetic field positions of B_1 and B_2 follow quasi-2D behaviour, which is expected in graphite.
3. The revised manuscript/SI now contains a complete description of the theoretically expected behaviour for the various thermodynamic and transport probes, including conductance, magnetization, MCE, thermopower and thermoconductance.

4. We revised the discussion on the condition for a double-peak structure to merge into a single peak feature.

Since we believe that we have fully addressed the points made by referee, we now kindly ask referee to reconsider our paper for publication.

List of minor changes

In addition to the changes listed below each referee comment, we also introduced a few minor changes to improve the readability. All the changes are highlighted by red color.

I. Manuscript

Page 2, line 26: We have revised the text to “in striking contrast to the single peak feature expected from Lifshitz-Kosevich theory.”

Page 2, line 27: We have added the text “the kernel term for”

Page 2, line 30: We have deleted the text “which involves the integral of energy multiplied by the temperature derivative of the Fermi-Dirac function convoluted with the Landau level density of states.”

Page 2, line 32: We have revised the text to “such as Lifshitz transition in heavy-fermion compounds.”

Page 2, line 46: We have added the text “wide”

Page 3, line 58: We have added the text “This result is in striking contract to the single peak feature predicted in LK theory for the quantum oscillations of specific heat, which is widely used in the literature [11–13] (see also”

Page 3, line 62: We have added the text “the double-peak structure in the oscillatory specific heat originates from the kernel term in the detailed functional form of the free electron theory expression for the specific heat [14]. A quantitative understanding of the double-peak structure is achieved by the comparison of a DOS model and the Slonczewski-Weiss-McCure (SWM) tight binding Hamiltonian for graphite [15, 16]. Using graphite as an example, we demonstrate that the double-peak structure”

Page 3, line 68: We have added the text “without any assumptions concerning the Landau index or Fermi energy shift”

Page 3, line 75: We have revised the section title from “II. Experimental results” to “RESULTS”

Page 3, line 76: We have added the subsection title “Experimental results”

Page 7, line 139: We have added the subsection title “Origin of double-peak structure”

Page 7, line 142: We have moved $-x^2$ into the bracket, and deleted the right part of Eq.1 “ $k_B^2 \int_{-\infty}^{\infty} D(E) \frac{x^2 e^x}{e^x + 1} dx$ ”.

Page 7, line 144: We have added the text “kernel term”

Page 7, line 148: We have corrected the text “double peak structures” to “double-peak structure”

Page 7, line 149: We have added the text “the kernel term”

Page 7, line 150: We have added the text “the kernel term”

Page 8, line 177: We have added the text “spin”

Page 9, Figure 3: We have added the text “kernel”

Page 10, line 198: We have added the text “kernel term”

Page 10, line 199: We have added the text “The peaks in Cel/T appear at certain magnetic field when the DOS peak is tuned to the maxima of the kernel term $-x^2 dF(x)/dx$ ”

Page 11, line 209: We have deleted the text “to”

Page 12, line 221: We have added the subsection title “Estimate of the dE/dB from the double-peak structure”

Page 12, line 242: We have added the subsection title “Estimate of the g -factor from the double-peak structure”

Page 14, line 275: We have added the subsection title “Double-peak structure in the Lifshitz transition”

Page 14, line 292: We have added the subsection title “Kernel term for different probes”

Page 16, line 308: We have revised the section title from “III. Conclusion” to “DISCUSSION”

Page 17, line 325: We have added the text “Sample#1 – Sample#3” and “0.18 mg, 0.23 mg”

Page 17, line 342: We have added the text “To measure the angle-dependence of the specific heat in magnetic field, a CERNOX resistive chip is mounted on a copper ring attached to an attocube rotator. On the back of the copper ring, a Hall probe allows to measure the angle with the magnetic field. The misalignment between the sample and the Hall probe is estimated to be within ± 2 degrees.”

Page 19, line 368: We have added the section “AUTHOR CONTRIBUTIONS”

Page 19, line 376: We have added the section “COMPETING INTERESTS”

Page 19, line 378: We have added the section “ADDITIONAL INFORMATION”

II. Supporting information

Page 18, line 212: We have added the text “DOS peak” and “kernel term”

Page 18, line 213: We have added the text “provided the DOS peak is symmetric (e.g. cusp-like DOS in Lifshitz transition). We validated this hypothesis by calculating the overlap integral versus energy, for different temperatures and width Γ , as the DOS peak passes through the Fermi energy”

Page 18, line 216: We have added the text “ $\Delta B dE/dB$ ”

Page 18, line 217: We have added the text “using a symmetric DOS peak”

Page 18, line 221: We have added the text “for a symmetric DOS peak are in good agreement with expected 4.8kB T regardless of the temperatures and the width Γ .”

Page 18, line 223: We have added the text “However, this is not exactly the case for a highly asymmetric DOS peak (e.g. DOS in Landau levels).”

Page 18, line 227: We have added the text “For this reason, we performed similar simulation using a highly asymmetric DOS peak”

Page 18, line 230: We have added the text “using an asymmetric DOS peak”

Page 19, Fig. S8: We have added the panel (a) and revised caption accordingly

I believe that Yang et al. have addressed the previous questions satisfactorily. However, I would like to offer several additional comments based on the new data and results:

1. There seems to be an inconsistency in the positions of the peaks in conductance and specific heat. According to the section titled “VI. Exact form for various thermodynamic probes,” the conductivity peak should occur at an energy of 0, while the specific heat peaks should be at approximately -1.6 and +1.6 K (as shown in Fig. 5). Nevertheless, in Fig. 1, which is also provided below for reference, the conductivity peak does not align with the midpoint of the two peaks in the specific heat.

2. Regarding reference 8, please clarify the identification of the negative and positive peaks (in the thermopower figure). It would be helpful to include this figure and provide clear identifications in the supplementary materials.

4. Could you shed some light on the reason behind the significant variation in double-peak positions among different samples, as depicted in Fig. S2 (also included below)?"

5. In Phys. Rev. B 68, 165408 (2003), the field-dependence of thermal conductivity in graphite was reported. However, unlike the findings of the current authors, the authors of that study did not report a double-peak structure in thermal conductivity, as depicted in Fig. 5.

6. The term "hidden" in the title appears peculiar. After reading the manuscript, what exactly is the double-peak structure hidden from? Based on the authors' explanation, the double-peak structure is not something unexpected or concealed; instead, it is a natural outcome rather than a surprising hidden finding in specific heat measurement.

Reviewer #2 (Remarks to the Author):

The authors carefully addressed my previous comments. The revised manuscript further clarifies the importance of the paper in this research area and stimulates future research; I recommend its publication in Nature Communications.

Reviewer #3 (Remarks to the Author):

The authors have addressed all comments from all referees satisfactorily. It is now acceptable for publication in NC.

Unveiling the double-peak structure of quantum oscillations in the specific heat

Zhuo Yang, Benoît Fauqué, Toshihiro Nomura, Takashi Shitaokoshi, Sunghoon Kim, Debanjan Chowdhury, Zuzana Pribulová, Jozef Kačmarčík, Alexandre Pourret, Georg Knebel, Dai Aoki, Thierry Klein, Duncan K. Maude, Christophe Marcenat and Yoshimitsu Kohama

REVIEWER COMMENTS

Reviewer #1 (Remarks to the Author):

Referee: I believe that Yang et al. have addressed the previous questions satisfactorily. However, I would like to offer several additional comments based on the new data and results:

Authors: We thank the referee for the positive appraisal of our work, and we highly appreciate the comments/suggestions made by the referee, which greatly helped us to improve the manuscript.

Referee: Comment 1) There seems to be an inconsistency in the positions of the peaks in conductance and specific heat. According to the section titled “VI. Exact form for various thermodynamic probes,” the conductivity peak should occur at an energy of 0, while the specific heat peaks should be at approximately -1.6 and +1.6 K (as shown in Fig. 5). Nevertheless, in Fig. 1, which is also provided below for reference, the conductivity peak does not align with the midpoint of the two peaks in the specific heat.

Authors: We thank the referee for this interesting point. We fully agree with referee on this statement.

We compared in Table R1 the conductivity peak positions (ΔR_{xx}) in our work and in the literature (*Phys. Rev. Lett.* 102, 166403 (2009)), as well as the midpoint of the double-peak structure in C_{el}/T . The conductivity peak positions in our work are almost identical to that in the literature (variation within 0.3% - 1%), while the midpoint of double peaks in specific heat is 3% - 4% different from the conductivity peaks. Thus, there is a reproducible tiny difference between the peak positions in conductivity and specific heat.

Except for the kernel term shown in manuscript Fig. 5, there are a few other factors that affects the peak positions, which can be responsible for this difference. The first one may be due to the fundamental differences between conductivity and specific heat.

Conductivity is a surface-sensitive technique that probes delocalized states, while the specific heat is a bulk-sensitive technique that probes both delocalized and localized states. Since the shape of the experimental data also depends on the DOS, the different sensitivity to the exact nature of the DOS may account for the different peak positions between conductivity and specific heat.

Moreover, conductivity is a transport phenomenon that also depends on the time of scattering. In the peculiar regime of the quantum limit, it has been shown theoretically that the shape of the last quantum oscillations depends on the nature of the dominant scattering process (*J. Phys. Chem. Solids* **10**, 254-276 (1958)). The difference between the peak position in conductivity and specific heat measurements could be then a consequence of the field dependence of the time of scattering.

In addition to these fundamental differences, it is worth noting that the samples were measured in completely different setups, therefore, the 3 - 4% deviation between different probes is reasonably within the experimental uncertainty (e. g. slight field inhomogeneity). The variation between different techniques was also observed in the Table I of the thermopower of graphite paper (*Phys. Rev. B* **3(4)**, 1148 (1971)).

Importantly, we believe that this tiny deviation does not affect the conclusion of our work, which is a single-peak feature in $MCE/\Delta R_{xx}$ and a double-peak structure in C_{el}/T . We have now clarified this point in manuscript line 296.

Table RI. Summarized magnetic field position of conductivity peak in both literature and this work, as well as the midpoint of double-peak structure in C_{el}/T . Note that the midpoints of C_{el}/T are taken from Table I in the manuscript.

	ΔR_{xx} in literature	ΔR_{xx} in this work	C_{el}/T in this work
1_e^-	7.63	7.55	7.87
1_e^+	6.84	6.77	7.05
1_h^-	3.71	3.7	3.86
1_h^+	3.48	3.5	3.61
2_e^-	3.03	3.04	3.17
2_e^+	2.94	2.95	3.08
Difference of ΔR_{xx} in ref and this work	0.3% - 1%		
Difference between ΔR_{xx} and C_{el}/T	3% -4%		

Changes:

I. Manuscript

Page 15, line 296: We revised the text “Although the shape of experimental data for different probes can be influenced by the $D(E)$ or other factors (e.g., scattering time for transport probes), it is the kernel term that determines the shape of the experimental data to be single- or double-peak feature.”

Referee: Comment 2) Regarding reference 8, please clarify the identification of the negative and positive peaks (in the thermopower figure). It would be helpful to include this figure and provide clear identifications in the supplementary materials.

Authors: We thank the referee for the interesting suggestion. We have included the thermopower figure from reference 8 in the **Sec. IX. A** in the supplementary materials. The negative and positive peaks are marked as blue and red arrows, respectively.

In addition to reference 8 (*Phys. Rev. B* **3(4)**, 1148 (1971)), we also included more recent thermopower data of natural graphite with identifications for the negative/positive peaks (*Nat. Phys.* **6(1)**, 26-29 (2010)).

Changes:

I. Manuscript

Page 15, line 301: We added text "(see Supplementary Sec. IX)"

Page 22, line 469: We added reference "[41] Z. Zhu, H. Yang, B. Fauque, Y. Kopelevich, and K. Behnia, Nernst effect and dimensionality in the quantum limit, *Nat. Phys.* **6**, 26 (2010)"

II. Supplementary information

Page 16, line 190: We added subsection "IX. A. Negative and positive peaks in thermopower of graphite"

Referee: Comment 3) Could you shed some light on the reason behind the significant variation in double-peak positions among different samples, as depicted in Fig. S2 (also included below)?"

Authors: We thank the referee for the question. The variation in double-peak positions among different samples is mainly attributed to the different measurement temperature and different Fermi energy among samples.

1. Different measurement temperature

As marked in **SI Fig. S2**, sample#1 and sample#2 were measured at $T = 0.5$ K, while sample#3 was measured at $T = 0.6$ K. Since the splitting of double-peak structure is temperature dependent, it is reasonable that the splitting of $1e^+$ level in sample#3 is larger than the one in sample#1. Note that the peak positions of Sample#1 and Sample#2 are identical since they were measured at the same temperature.

We added a sentence to the caption of **SI Fig. S2** to emphasize the measurement temperature for each sample.

2. Different Fermi energy among samples

The Fermi energy can be slightly different among samples due to different defect density, which determines the occupation of the electron and hole pockets. The variation of the Fermi energy influences the crossing field between Landau level and Fermi level, which subsequently causes the discrepancy in double-peak positions among different samples.

We evaluated the discrepancy of the midpoint of the double-peak structures for different samples to be 0.5% - 1%, which is a relatively small difference. Therefore, we believe that the main contribution to the variations among samples is the different measurement temperatures.

Changes:

II. Supplementary information

Page 5, Figure S2: We have added sentence to the caption “Sample#1 and Sample#2 were measured at $T = 0.5$ K. Sample#3 was measured at $T = 0.6$ K.”

Referee: Comment 4) In *Phys. Rev. B* **68**, 165408 (2003), the field-dependence of thermal conductivity in graphite was reported. However, unlike the findings of the current authors, the authors of that study did not report a double-peak structure in thermal conductivity, as depicted in Fig. 5.

Authors: We thank the referee for the comment. We believe there are mainly two reasons that prevent the authors from observing double-peak structure in thermal conductivity, namely, (1) too high measurement temperature; (2) low sample quality.

1. Too high measurement temperature

The lowest measurement temperature of thermal conductivity in this reference is 5 K (also included as Fig. R1(a)), which is too high to observe the double-peak structure. At this temperature range ($T > 4$ K), we could not observe double-peak structure in specific heat either, as shown in Fig. R1(b). The double-peak structure in specific heat is most clearly observed at $T < 1$ K, as seen in manuscript Fig. 2(a).

We note that the prediction of the thermal conductivity in our manuscript is about electronic contribution. However, the thermal conductivity data generally consists of electron and phonon contributions. In semi-metals with relatively high measurement temperature, the quantum oscillations in thermal conductivity are possibly originated from the oscillations of the phononic contribution as a consequence of the modulation of the phonon mean-free path by the quantized electrons due to the electron-phonon scattering (*Phys. Rev. X* **12**,031023(2022)). Considering the large magnetoresistance of graphite, the electronic contribution to thermal conductivity is extremely small even at lowest temperature. At $T > 5$ K and in the presence of a magnetic field, the thermal conductivity for graphite is very likely to be dominated by the phonon contribution, and thus will not show double-peak structure.

2. Low sample quality

The thermal conductivity measurements have been done in highly oriented pyrolytic graphite (HOPG), which is notoriously known to have a much broader Landau level width than natural graphite (see SI Sec. V). In HOPG samples, we did not detect, even at the lowest temperature, the double-peak structure in specific heat. It is thus not surprising to not observe it in thermal conductivity.

The condition to observe a double-peak structure is that the line width of the DOS peak has to be much smaller than the splitting of the kernel term ($4.8k_B T$). As shown in

SI. Sec. V, the Landau level broadening Γ of HOPG is 0.55-0.73 meV, which is generally larger than $4.8k_B T$ at low temperature (e.g., at 1 K, $4.8k_B T = 0.4$ meV). For this reason, this condition cannot be fulfilled in HOPG.

Therefore, in order to detect the double-peak structure in the electronic contribution of the thermal conductivity of graphite, one has to measure it in a natural graphite sample (with the narrowest Landau level broadening Γ) at around 100 mK and carefully subtract the phonon contribution.

Fig. R1 (a) Digitalized thermal conductivity data of graphite from reference *Phys. Rev. B* **68**, 165408 (2003). (b) Field sweep of specific heat C/T at $T > 4$ K. The magnetic field position of spin-splitting for the Landau levels are marked as coloured transparent lines.

Referee: Comment 5) The term "hidden" in the title appears peculiar. After reading the manuscript, what exactly is the double-peak structure hidden from? Based on the authors' explanation, the double-peak structure is not something unexpected or concealed; instead, it is a natural outcome rather than a surprising hidden finding in specific heat measurement.

Authors: We thank the referee for the comment. Hidden, referred to the fact that the double-peak structure was not previously observed/identified in experiment. We have now deleted the term "hidden" in the title.

We fully agree with the referee that the double-peak structure is a natural outcome when the exact form of specific heat is considered. However, this feature has long been overlooked, because researchers tend to use the well-known formula $C_{el} = \pi D(E_F)k_B^2 T/3$, which actually suppressed the double-peak feature.

In the aspect of experiments, the condition to observe double-peak structures is rather rigorous, it requires extremely high quality of the sample, ultra-low temperature, as well as high precision of temperature calibration. For these reasons, there are almost no experimental reports of these double-peak features.

As 'hidden' was not used in the main text, we have simply removed it from the title following the suggestions of the referee.

Deletion:

I. Manuscript

Page 1, line 1: We deleted the term 'hidden' in the title.

II. Supplementary information

Page 1, line 1: We deleted the term 'hidden' in the title.

Changes:

I. Manuscript

Page 1, line 1: We added the term 'the' in the title.

II. Supplementary information

Page 1, line 1: We added the term 'the' in the title.

Authors: We thank again the referee for constructive suggestions. Since we believe that we have fully addressed the points made by the referee, we now kindly ask the referee to reconsider our paper for publication.

Reviewer #2 (Remarks to the Author):

Referee: The authors carefully addressed my previous comments. The revised manuscript further clarifies the importance of the paper in this research area and stimulates future research; I recommend its publication in Nature Communications.

Authors: We thank the referee for the positive appraisal of our work, and we appreciate the comments/suggestions made by the referee, which greatly helped us to improve the manuscript.

Reviewer #3 (Remarks to the Author):

Referee: The authors have addressed all comments from all referees satisfactorily. It is now acceptable for publication in NC.

Authors: We thank the referee for the positive appraisal of our work, and we appreciate the comments/suggestions made by the referee, which greatly helped us to improve the manuscript.

List of minor changes

In addition to the changes listed below each referee comment, we also introduced a few minor changes to improve the readability. All the changes are highlighted by red color.

I. Manuscript

Page 5, Figure 1: We have aligned the subscript and superscript of the notation for Landau levels. An example is given as below. This minor change was also applied to **Figure 2 – 4**.

$$1_e^+ \longrightarrow 1_e^+$$

Page 15, Figure 5: We have added a sentence to the caption “All the curves were calculated at $T = 0.5$ K”.

II. Supplementary information

Page 5, Figure S2: We have aligned the subscript and superscript of the notation for Landau levels. This minor change was also applied to **Figure S3**.

Page 12, line 149: We have added the term “or transport”